# Implicit Probabilistic Reasoning Does Not Reflect Explicit Answers in Large Language Models

**Manuel Mondal**  *manuel.mondal@unifr.ch*
*University of Fribourg, Switzerland*

**Ljiljana Dolamic**  *ljiljana.dolamic@armasuisse.ch*
*armasuisse S+T, Switzerland*

**Gérôme Bovet**  *gerome.bovet@armasuisse.ch*
*armasuisse S+T, Switzerland*

**Philippe Cudré-Mauroux**  *philippe.cudre-mauroux@unifr.ch*
*University of Fribourg, Switzerland*

**Julien Audiffren**  *julien.audiffren@unifr.ch*
*University of Fribourg, Switzerland*

**Reviewed on OpenReview:** *https://openreview.net/forum?id=HaaAY4ZXPa*

## Abstract

The handling of probabilities in the form of uncertainty or partial information is an essential task for LLMs in many settings and applications. A common approach to evaluate an LLM's probabilistic reasoning capabilities is to assess its ability to answer questions pertaining to probability through the use of multiple-choice questions (MCQs). However, this paradigm, which we refer to as *explicit probabilistic reasoning*, has been shown in the literature to yield significant limitations (e.g., sensitivity to answer ordering). In this work, we introduce an alternative approach, named *implicit probabilistic reasoning*, which evaluates the models' ability to integrate probabilistic reasoning into their text generation process. To achieve this, we rephrase MCQs as text-completion scenarios with a determined set of outcomes and compare the model's next-token probability assignments to the true likelihood of the outcomes. In line with previous work, we find that models exhibit solid performance in their explicit probabilistic reasoning (i.e., answers to MCQs). However, during text completion (i.e., implicit probabilistic reasoning), where the same information must be taken into account to generate text, the models' predictions often significantly diverge from the known ground truth. For instance, our evaluation method reveals that implicit probabilistic reasoning is improperly influenced by many factors, such as independent prior events, partial observations about a result, or statistical background information. All of these issues can cause erroneous results to be produced in text generation, which are not detected by conventional MCQ-based evaluations.

## 1 Introduction

In recent years, Large Language Models (LLMs) have gained significant traction in the research community and the public at large (Zhao et al., 2023; Chang et al., 2024). At their core, LLMs are statistical models of languages that predict, for a given context, a probability distribution over their vocabulary for the occurrence of the next token in a sequence (Vaswani et al., 2017). Despite this simplicity, a wide array of earlier research has noted that their sophisticated use of natural language is impressive (Chang et al., 2024; Hu & Levy, 2023) and some studies have claimed that LLMs have become more than just statistical models,

gaining emergent abilities due to their massive training sets and architecture sizes (Bubeck et al., 2023; Wei et al., 2022) – including reasoning and probability manipulation (Kıcıman et al., 2023; Nafar et al., 2024). However, this fact is debated in the research community. While recent LLMs perform well on advanced benchmarks (Srivastava et al., 2023; Phan et al., 2025), they can also be tricked by simple questions and adversarial prompt modifications (Xu et al., 2023; Zou et al., 2023). This has raised the question of whether LLMs indeed have an aptitude for reasoning, or whether these observations are an illusion that emerges from data memorization.

At the heart of these debates is the problem of evaluating LLMs. One of the most common ways to measure their performance is through prompting (see e.g., Brown et al. (2020)), and most benchmarks are collections of questions and answers (Hendrycks et al., 2021; Srivastava et al., 2023; Liang et al., 2023) where the LLMs are prompted with a question and the resulting answer is compared to the known ground truth. By nature, this evaluation method is vulnerable to data contamination, where the LLMs observed part of the evaluation set (or close phrasings thereof) during training, a problem exacerbated by the fact that the training datasets of most LLMs are generally not accessible to the research community (Deng et al., 2024). Furthermore, since the evaluation of open-ended answers is quite complex and resource-consuming (Frieder et al., 2024), the questions in these benchmarks most often boil down to Multiple Choice Questions (MCQs) (Liang et al., 2023), where LLMs are asked to choose from a finite set of answers. The evaluation via MCQs has been shown to suffer from a variety of biases (Pezeshkpour & Hruschka, 2024; Wang et al., 2024a). For instance, the answer of an LLM to an MCQ can be strongly influenced by the ordering of the answers (Pezeshkpour & Hruschka, 2024). These problems hint at the fact that the formatting of the questions may be as important to the LLM's performance as the question itself, throwing doubt on the interpretation of the results.

Recently, multiple works have investigated the evaluation of LLMs, e.g., by probing their internal representations (Alain & Bengio, 2017; Azaria & Mitchell, 2023; CH-Wang et al., 2024), or by analyzing their next-token distributions (Feng et al., 2024; Slobodkin et al., 2023), in particular to study their confidence and calibration (Kuhn et al., 2023; Lin et al., 2022; Kadavath et al., 2022). Notably, Hu & Levy (2023) have compared the merits of evaluating LLMs with direct text completion. While their experiments focus on the linguistic capabilities of the LLMs and their knowledge of the English language, their work suggests a distinction between a model's performance (its behavior when prompted with questions) and a model's competence (the information contained in an LLM's string probability distribution). In line with these results, we argue that evaluating LLMs by exclusively assessing their answers to carefully crafted questions provides an incomplete assessment of LLMs' abilities.

In this work, we focus on the LLMs' ability to leverage probabilistic information. Probabilistic information is present in many documents or prompts, and its proper integration can be key to producing an adequate answer. For instance, a straightforward application of Bayes' theorem shows that the statistical prevalence of a disease strongly influences its corresponding probability of diagnosis, see e.g., Leeflang et al. (2009) for some examples. We examine such scenarios in the case study in Section 3.2 and our experimental results in Section 4. The ability of LLMs to manipulate probabilities has been studied in the past through conventional MCQ (Wang et al., 2024b; Hendrycks et al., 2021; Nafar et al., 2024; Srivastava et al., 2023), hinting at their promising abilities. We introduce in this paper a different paradigm to evaluate the LLMs' ability to manipulate uncertainty. In this paradigm, we evaluate the models' *implicit* probabilistic reasoning capability, relying on their next-token prediction (LLMs' elementary unit of computation), forgoing the conventional *explicit* question/answer framework.

To illustrate these two evaluation paradigms, consider the example of the roll of two fair dice. To measure a model's explicit handling of probabilities, we query the model with a multiple-choice question such as: "Two fair six-sided dice are cast. What is the probability that the sum of their faces is equal to 7? A) $1/2$, B) $1/4$, C) $1/6$, D) $1/8$". In contrast, we measure a model's implicit handling of probabilities by its prediction of next-token probabilities in a text completion phrasing of an equivalent scenario: "Two fair six-sided dice are cast. The sum of their faces is equal to <tokens>", and in particular the probability of the token "7" – which is the most likely outcome. As next-token probability prediction is the fundamental mechanism which determines an LLMs text continuation, this latter approach allows us to directly evaluate whether probabilistic information is properly integrated into the text-generation process. Ideally, models would assign the true likelihood to each outcome and thereby generate text aligned with the known probabilities or at least

favor the most likely outcome. Conversely, significant differences between this and the answers produced by models when queried with MCQs would hint at the limits of current benchmarks.

More generally, we construct a set of evaluation tasks where we present an LLM with some context (e.g., a dialog with a user) describing a probabilistic scenario that has multiple possible outcomes (e.g., the statistical likelihood of several pathologies) and use the LLM's text completion prediction to simulate its resolution (e.g., which diagnosis is favored by the LLM). Under the assumption that LLMs are able to manipulate probabilities, we expect the model to generate text in accordance with the given information, i.e., to produce a probability distribution over the outcomes that is aligned with the true distribution, or at least that the two distributions share the same Maximum Likelihood (ML). This latter point is important, given that the sampling of the next-token during the autoregressive text-generation will select the token with the predicted ML more frequently (particularly for temperatures $t < 1$). If the MLs of the two distributions were not properly aligned, the LLM would incorrectly favor a less likely outcome, which would, in our aforementioned example, lead to the overdiagnosis of rare pathologies. In line with the "show, don't tell" adage, this approach emphasizes the model's implicit reasoning ability, instead of their explicitly stated answer to a predetermined MCQ (Section 3). We argue that this approach is complementary to a measurement of an LLMs' ability to answer explicit questions about probabilities, by additionally measuring the model's ability to integrate probabilistic information from its context into its prediction about the outcome of a scenario.

We apply this new paradigm in a wide range of scenarios (see Section 3.5) and make several key observations. First, while the LLMs perform well when handling probabilities explicitly in a multiple-choice question-answering setting, we observe that their implicit ability to do so performs significantly worse, indicating that the good performance of LLMs on probabilistic questions in MCQs is not representative of their abilities at handling uncertainty in their generated text. Second, their implicit handling of probabilities allocates probability mass inconsistently, which may result in the LLM favoring unlikely outcomes. Third, these analyses also highlight that LLMs' probabilistic predictions are affected inappropriately when presented with (i) partial evidence and (ii) the occurrence of unrelated prior events (see Section 4). These results hint at the limitations of current evaluation methods and suggest that additional research is needed.

## 2 Related work

**LLMs and probability.** Previous studies examined the aptitude of LLMs to handle problems involving probabilities. Recent contributions include Nafar et al. (2024), which studied the reasoning capabilities of LLMs on text that contains explicit probability values, and pointed to reasonable performance by LLMs. In subsequent work, the authors use LLMs as a source for extracting meaningful probabilistic estimates for events to parametrize Bayesian networks and reason over these Nafar et al. (2025). Likewise, modern LLMs achieve solid performance on MCQ queries about the explicit handling of probabilities contained in common benchmarks (Hendrycks et al., 2021; Wang et al., 2024b). Similarly the authors in Paruchuri et al. (2024) investigate how LLMs handle statistical distributions explicitly, for instance by asking models to estimate the percentile of a sample within a distribution. However, other recent works, such as Jin et al. (2023); Frieder et al. (2024) hinted at more disappointing results, in particular for more advanced causal and mathematical problems. Approaches to address shortcomings in probabilistic reasoning have recently been explored Feng et al. (2025) but mainly rely on using the LLMs in an abductive reasoning step to externalize the probabilistic inference to a secondary Bayesian model. Compared to these works, the probability problems involved in our experiments are significantly easier, such as the throw of a die. Moreover, all the aforementioned studies evaluate the explicit probabilistic reasoning ability of LLMs, whereas we investigate their implicit ability (see Section 3), which led to significantly different observations.

**Investigating LLM evaluation.** Previous work also questioned the evaluation of LLMs and scrutinized their flaws. In particular, MCQs – one of the most prevalent types of evaluation (Hendrycks et al., 2021; Srivastava et al., 2023; Liang et al., 2023) – faced many criticisms. For instance, Pezeshkpour & Hruschka (2024) showed that merely reordering the options of MCQ can lead to double-digit performance gaps across multiple benchmarks, while Alzahrani et al. (2024) additionally showed that a similar phenomenon can be observed by changing the numbering scheme of the provided answers. In addition, Balepur et al. (2024) found that LLMs can outperform random guessing even when presented with choices-only prompts, without

the questions themselves. This hints at the model's surprising ability to exploit dynamics between answer options and infer (or recall) the original question. Other works pointed to the problem of data contamination, where the LLMs are shown to have been exposed during training or fine-tuning to the evaluation data used in common benchmarks (Deng et al., 2024; Balloccu et al., 2024).

**Model confidence and calibration** Other works have focused on quantifying model confidence and calibration. Kuhn et al. (2023) use next-token likelihoods to represent lexical confidence. However, their solution to this problem (measuring semantic uncertainty of different text-completion candidates) is not directly applicable to cases where multiple outcomes are valid, yet semantically different (e.g., the outcome of a die roll). Works such as Lin et al. (2022) and Kadavath et al. (2022) propose supervised approaches to extract and reduce the model's explicitly stated confidence in its answer. They achieve this by fine-tuning LLMs to state their confidence in the generated text, or by sampling multiple text-generations and letting the model judge these samples. Other approaches, such as Jiang et al. (2021); Achiam et al. (2023); Zheng et al. (2024); Kadavath et al. (2022), measure the model's assigned probability to its chosen answer in a multiple-choice question and consider this the model's confidence in its answer. Jiang et al. (2021) propose fine-tuning and post-hoc calibration approaches to calibrate the model's confidence in its answers (targeting a high probability when the answer is correct and low when it is not). Our proposed evaluation approach differs in as much as we do not consider the models' predicted probability assignments on their explicit answer to questions, but rather the completion of scenarios describing this question.

**LLM next-token probability assessments.** Hu & Levy (2023) highlight the discrepancy between the direct completion of a text and prompting LLMs to complete the text, with particular emphasis on their mastery of the English language. While our measurements of implicit and explicit probability reasoning follow a similar dichotomy, the scope of our study (scenarios with uncertainty and multiple outcomes) differs significantly, and we investigate the details of the LLM's implicit biases and errors. In another work, Qiu et al. (2025) evaluate the ability of LLM-based assistants to update their beliefs on user preferences in multi-round conversations. The authors retrieve the LLM's beliefs (e.g., by asking the model to state an explicit probability distribution over the user's preferences) and use this distribution to compute the assistant's recommendation. The models are then evaluated on the accuracy of this downstream recommendation (with a single correct answer). Our approach to assess the implicit reasoning abilities is very different, as we focus our evaluation on the predicted probability distribution directly, instead of asking the model to state them, and we consider cases with a plurality of possible outcomes. Furthermore, we also examine additional probabilistic computation and tasks, such as the ability to integrate (or discard) dependent (or independent) prior results and consider different probability distributions.

## 3 Methodology

### 3.1 Explicit and Implicit probabilistic reasoning

**Explicit probabilistic reasoning.** We evaluate a model's explicit probabilistic reasoning ability based on the answer it gives to a question requiring the handling of probabilities. The model's answer is commonly determined from its output by sampling a response token, by parsing choices from some generated text (e.g., from "My answer is: D"), or, as in our experiments, by computing the probability of the LLM selecting the correct answer, optionally after chain-of-thought reasoning tokens (see Figure 1, left).

LLM probabilistic reasoning capabilities are commonly assessed by measuring their ability to state the correct answer in multiple-choice questions (MCQs) (Wang et al., 2024b; Hendrycks et al., 2021; Nafar et al., 2024; Srivastava et al., 2023). In a typical setup, the model receives a question (sometimes along with some background information) and a set of options (generally four to ten) with a single correct answer. For instance, the following MCQ could be used to evaluate explicit probabilistic reasoning:

Explicit probabilistic reasoning | Implicit probabilistic reasoning

Context      Prob. of interest      Context      Prob. of interest

What is the probability that two fair six-sided dice land on 7? A) $\frac{1}{36}$ B) $\frac{1}{3}$ C) $\frac{1}{6}$ D) $\frac{1}{12}$ $\xrightarrow{\text{LLM}}$ $\mathbb{P}_{\text{LLM}}\big(C|\text{context}\big)$ | Two fair six-sided dice land on <tokens> $\xrightarrow{\text{LLM}}$ $\mathbb{P}_{\text{LLM}}\left(\begin{pmatrix} 2 \\ 3 \\ .. \\ 11 \\ 12 \end{pmatrix} |\text{context}\right)$

Figure 1: Illustration of implicit and explicit reasoning with probabilities, for a scenario with two six-sided dice. Examples of detailed prompts can be found in Appendix A.

> QUESTION. What is the probability that two fair six-sided die rolls sum to 7?
>
> A) $1/36$  B) $1/3$  C) $1/6$  D) $1/12$

Despite their wide adoption, such evaluations exhibit several well-documented limitations, such as sensitivity to the ordering of the answer choices (Pezeshkpour & Hruschka, 2024). Moreover, selecting an answer out of a set of provided options is a discriminative task that differs from the generative way in which LLMs are commonly used.

**Implicit probabilistic reasoning.** In contrast, the evaluation of implicit probabilistic reasoning we propose assesses LLMs at the level of their fundamental computational unit: the probability distribution they assign to the next token, given an input sequence. In this framework, each multiple-choice question is reformulated as a text-completion task: the model receives a scenario and we assess the probability distribution it assigns to the set of possible outcomes (see Figure 1, right). We extract the raw logits produced by the LLM and convert them into a probability distribution over the next token (or a specific token sequence) by applying a softmax operation, denoted $\mathbb{P}_{\text{LLM}}(\cdot|\text{context})$. When the answer is composed of multiple tokens, (e.g., depression is tokenized as de|pression by the Llama tokenizer, see Section 3.2 below) the probability of the answer is computed as the joint probability of the sequence of tokens.

For instance, in the previous die roll example, this approach will compute the likelihood of obtaining each outcome '2', '3', ..., '12'. As this probability distribution will be used to continue the scenario, the model ought to weigh the most likely outcome ('7') the highest, if the model is expected to generate text that is coherent with true probabilities. Comparing this distribution to the known ground-truth lets us quantify how closely the model's implicit predictions align with the true probability of the events, which may have a significant impact on a model's response. Indeed, LLMs generate text by sampling from this distribution, and its shape determines the ensuing text generation. Consequently, an erroneous (e.g., skewed, biased) probability distribution can yield an inappropriate next-token choice, which can propagate to the entire text generated downstream. For instance, this may result in the LLM favoring an unlikely medical diagnosis and then providing a false rationalization, as illustrated by the case study in the following subsection.

## 3.2 Case study

We illustrate the contrast between the two evaluation paradigms (of explicit and implicit probabilistic reasoning performance) by comparing the same language model (here: *Llama-3.3-70B*[1]) to answer a medical query (see the Statistics scenario in Section 3.5). In this setting, we provide the model with the same statistical background information and it is queried: (i) in the form of a multiple-choice question (explicitly stating its answer) and (ii) as an open-ended user dialogue, where the model completes the response (implicitly applying probabilistic reasoning to generate a text continuation). Importantly, in both settings, the model receives the same prevalence statistics, which must be combined to answer the queries – see Figure 2.

In the (explicit) MCQ-setting, the problem is formulated as a multiple-choice question (see Figure 3a), where the model is queried about a subject's probability of suffering from one of the mentioned medical conditions. In order to answer this question, the model must perform a single-step Bayesian probability computation

---

[1]Similar behavior occurs across other models, as shown by the results in Section 4.

```
A study reported the prevalence of mental health conditions among hospital healthcare workers employed in
surgical wards:
- burnout:  8%
- anxiety:  13%
- depression:  7%
Among hospital healthcare workers who did not work in surgical wards, were:
- burnout:  16%
- anxiety:  10%
- depression:  5%
Overall, 18% of healthcare workers were employed in surgical wards.
Sam is a healthcare worker in a hospital.
```

Figure 2: The model is provided with identical background statistics on a medical condition. Note that given the provided information, the likelihood that Sam suffers from anxiety is $18\% \times 13\% + 82\% \times 10\% \approx 11\%$.

```
System:  <MCQ formatting instructions>
User:  <Statistical background> Sam is a
healthcare worker in a hospital.  Question:
Based on this data, what is the probability that
Sam suffers from anxiety?  Answers:
[[A]] 11%
[[B]] 10%
[[C]] 63%
[[D]] 13%
[[E]] 72%
Assistant:  [[A^{99.96%}]] 11%
To find the probability that Sam suffers from
anxiety, we need to calculate the overall
probability of anxiety among all healthcare
workers, taking into account those who work in
surgical wards and those who do not.
Let's denote:
- P(S) as the probability of working in a
surgical ward, which is 18% or 0.18.  - P(NS)
as the probability of not working in a surgical
ward, which is 100% - 18% = 82% or 0.82.
[...]
Converting this back to a percentage:  P(A) ≈ 11%
Therefore, the probability that Sam suffers from
anxiety is approximately 11%.
```

```
User:  <Statistical background>
Assistant:  Based on this data, I conclude that
Sam is most likely to suffer from anxiety^{99.08%},
as it has the highest prevalence rate among
the given mental health conditions, at 13% for
those in surgical wards and 10% for those not
in surgical wards.  However, without knowing
whether Sam works in a surgical ward or not, I
can calculate the overall probability of each
condition for Sam.
To find the overall probability, I need to
calculate the weighted average based on the
percentage of healthcare workers in surgical
wards (18%) and those not in surgical wards
(82%).
- Burnout:  (0.18 * 8%) + (0.82 * 16%) = 1.44% +
13.12% = 14.56%
- Anxiety:  (0.18 * 13%) + (0.82 * 10%) = 2.34% +
8.2% = 10.54%
- Depression:  (0.18 * 7%) + (0.82 * 5%) = 1.26%
+ 4.1% = 5.36%
Given these overall probabilities, anxiety still
has the highest prevalence rate at 10.54%.
Therefore, Sam is most likely to suffer from
anxiety.
```

(a) Query and output in the explicit probabilistic reasoning setting. The model is able to select the correct answer with a probability of 99.96% and produces a coherent justification afterwards.

(b) Implicit probabilistic reasoning performance. The diagnosis "anxiety" is predicted with a probability of 99.08%, despite being less likely than burnout, given these statistics. The ensuing justification is self-contradictory.

Figure 3: Evaluation prompts and outputs for the explicit (left) and implicit (right) probabilistic reasoning evaluation paradigms. The prompt is displayed in light red, the model's answer in dark red.

to update its prediction by taking into account (i) the prevalence among healthcare professionals, (ii) the prevalence among surgical ward staff, and (iii) the proportion of hospital healthcare professionals working in surgical wards. In our experiments, the model is able to use the information provided in the system prompt to answer the question correctly (choice [[A]]) with a probability of 99.96%. Interestingly, the model also provides a coherent justification for its choice, consistent with the provided background information.

In the implicit reasoning setting, the model is also given the population statistics. The beginning of an answer is then provided to the LLM, ending on the last token before the diagnosis (see Figure 3b, text in light red). Letting the model generate the rest of the answer will disproportionally favor the "anxiety" diagnosis: indeed, the model places a 99.08% probability mass on "anxiety" (and totaling less than 1%

on "burnout" and "depression"). This is in stark contrast with the likelihood of the diagnoses, which are ~11% for "anxiety" and ~15% for "burnout" – with "burnout" being a more likely diagnosis according to the provided statistics. Furthermore, in some cases, this choice has a downstream effect on the LLM's answer, as the model sticks to its initial prediction, citing the numbers it ignored, and providing the user with an answer that seems evidence-based yet is probabilistically unsound (see Figure 3b text in dark red).

**The importance of measuring implicit probabilistic reasoning abilities.** By evaluating the model's next-token probability assignment, this approach measures the fundamental mechanism by which LLMs generate text. In a Bayesian sense, this probability assignment could be interpreted as a credence or a model's belief about the outcome of a scenario. In particular, this framework makes possible to measure the impact of how probability assignments are impacted by additional information (e.g., a prior die roll, or partial observations). We argue that generating text in accordance with a known probability reveals a model's implicit probabilistic reasoning ability and should be measured.

Indeed, instead of asking the model what information it knows, we measure how well the model is able to integrate this information (e.g., the statistical likelihood of a diagnosis) into its predicted distribution about the likely outcomes of a scenario. As illustrated by the case study, a biased implicit reasoning process can lead to a skewed probability distribution over the possible outcomes, which can then lead to incorrect conclusions and misleading justifications. Furthermore, the model's explicitly stated answer is not necessarily indicative of a model's implicit prediction about the outcome, as the model appears able to correctly answer an MCQ without being able to integrate the information into next-token probabilities. Importantly, this fundamental discrepancy, as well as the biases of the implicit probabilistic reasoning process, are not specific to this case study, as it is reflected in many of our experimental results (see Section 4).

### 3.3 Models

To ensure the generality of our findings, we examine a broad set of models, covering a wide range of open-weight model providers: Qwen3-30B-A3B-Instruct-2507 (Q3-30B for short) (Yang et al., 2025), Qwen3-235B-A22B-Instruct-2507 (Q3-235B), Qwen2.5-72B-Instruct (QW-72B) (Yang et al., 2024), DeepSeek-R1-Distill-Qwen-32B (R1-32B) (Liu et al., 2024), DeepSeek-R1-Distill-Llama-70B (R1-70B), Llama-3.3-70B-Instruct (L3-70B) (Grattafiori et al., 2024), Mistral-Small-24B-Instruct-2501 (MI-24B), Mistral-Large-Instruct-2411 (MI-123B) (Jiang et al., 2023), gemma-3-27b-it (G3-27B) (Kamath et al., 2025), phi-4 (P4-14B) (Abdin et al., 2024). This selection aims at reflecting the model diversity and state of the art as of late 2025 at multiple-choice question answering and includes various model sizes (from 14B to 123B), architectures (dense and mixture-of-experts), as well as standard and reasoning-tuned models. We exclusively examine open-weight LLMs in order to analyze their next token distribution.

### 3.4 Evaluation setup

To measure the models' explicit probabilistic reasoning abilities, we evaluate the models first in the conventional benchmarking setup: they are prompted with the scenario's description, followed by a multiple-choice question. The model is instructed to format its final response within square brackets (see e.g., Figure 3a) and we retrieve the likelihood of the correct answer therein. The probability of the (only) correct response is then retrieved from this structure as the model's explicit answer to the MCQ.

To measure the models' implicit probabilistic reasoning performance, the models are presented with a rephrasing of the same scenario, where the prompts end on the token before the statement of the outcome. The probability distribution over the possible outcome tokens of the models is then computed and normalized. The probabilities assigned to the outcomes can be analyzed on two levels: the probability of a single outcome (such as the true maximum likelihood) and the distribution over all outcomes. The former allows for a one-by-one comparison between implicit and explicit probabilistic reasoning performance (see Section 4.1) and, in particular, permits checking whether the maximum likelihood is correctly favored. The latter allows for measuring the quality of how well the revealed implicit distribution matches the known ground truth, using three different metrics: the Chebyshev distance, the Manhattan distance, and the Kullback-Leibler divergence (see Section 4.2). While the last two metrics measure the total difference between the two dis-

tributions, the Chebyshev distance represents the maximal difference between the distributions and is thus particularly representative of the bias that models can have towards or against a specific outcome.

In both settings, the prompts are formatted specifically according to each model's chat template (e.g., by adding the special "user/assistant" tokens). To ensure consistent evaluation across models, we omit temperature-based probability rescaling and measure each model's raw next-token probability scores (i.e., assuming $t = 1$). As temperature does not affect the models' maximum likelihood prediction, this setting does not impact our empirical findings comparing implicit and explicit probabilistic reasoning. We use each model's weight precision as specified in their configuration file on Hugging Face (Wolf et al., 2019).

### 3.5 Tasks

To examine the aptitude of LLMs to handle uncertainty explicitly and implicitly, we set up five families of evaluation scenarios. Each LLM is evaluated on multiple combinations of scenarios, variants, and parameters (see Table 1). Scenarios are crafted to challenge the model in multiple ways: (i) identify which part of the provided information should affect the outcome (e.g., distinguish between dependent and independent prior events) and (ii) integrate probabilistic information revealed in the context into their prediction (e.g., updating their predictions given partial observation of the outcomes). Each evaluation prompt is formulated as an explicit multiple-choice question and as an implicit text-completion phrasing. Each family of scenarios is briefly described below and we refer the reader to Appendix A for detailed descriptions of the scenarios, configurations, and examples of prompts. Table 1 offers a summary of the different experimental setups.

**Dice Scenario.** In the first family (Dice), we construct prompts involving random dice rolls. To ensure that we cover a diverse spectrum of distribution shapes, the models are prompted with a varying number of dice and faces per die. These scenarios also include several variants, such as repeated rolls with a known prior outcome. In this setting, the context contains the result of a previous roll of the dice, and the model is asked to either realize a new independent roll (independent setting) or to sum both rolls (dependent setting). We also explore the impact of partial observations (e.g., "the result is greater than 3") to assess the models' ability to integrate incomplete information.

**Coins Scenario.** The second set of scenarios (Coins) focuses on random coin flips. There, we manipulate both the number of coins as well as explicit biases towards certain coin faces (e.g., "Heads is two times more likely than Tails") to probe how models respond to skewed distributions.

**Preference and Choice Scenarios.** The third family of scenarios (Preference) measures the impact of the outcome labels (e.g., "Left" vs. "Right") on the LLMs' distributional prediction. The fourth group of scenarios (Choice) studies a more abstract setting, where the labels of the outcomes are letters ('A', 'B', 'C', . . . ). These scenarios assess the models' handling of abstract or arbitrary choices, such as those encountered in multiple-choice questions, where labels have been shown to impact performance (Alzahrani et al., 2024).

**Statistics Scenario.** In a final setting (Statistics), we prompt models with scenarios including population statistics, similarly to the Case Study (Section 3.2). These statistics describe the prevalence of medical conditions in two populations, and the models are required to make use of this information to predict a diagnosis. We generated this synthetic evaluation dataset by randomly sampling (i) prevalence values of the diseases among the two populations, (ii) the relative size of the populations, and (iii) the name of the subject and hospital wards.

### 3.6 Probability of alternative answers

Importantly, as this second framework relies on evaluating the distribution of next-token predictions for a specified collection of tokens (e.g., '2', '3', '4'), we verified whether the LLMs' answers elicited unexpected continuation tokens (e.g., "two"). To measure this possibility, we defined in each scenario the set of valid continuation tokens (e.g., numbers 1-6 for the outcome of a six-sided die roll) and consider all other tokens as invalid continuations. Such discarded answers include, but are not limited to, syntactically valid continuations that can distort the examined probability distributions. For instance, the die roll scenario could be

| Scenario | Variants | Parameters |
|---|---|---|
| Dice | Single Roll
Repeated (Independent, Dependent)
Observations | Number of Dice, Number of Faces
Number of Dice, Number of Faces, Result previous roll
Number of Dice, Number of Faces, Observation(s) |
| Coins | Single Flip
Repeated (Independent, Dependent) | Number of Coins, Heads or Tails, Bias
Number of Coins, Heads or Tails, Bias, Result previous flip |
| Preference | Single Selection
Repeated (Independent) | Option 1, Option 2, Bias
Option 1, Option 2, Bias, Previous selection |
| Choice | Single Choice
Repeated (Independent) | Number of Options
Number of Options, Previous choice |
| Statistics | Simple | Prevalence rates of the different conditions, Population sizes
Names (of subjects and wards) |

Table 1: Summary of the scenarios, variants, and parameters.

Table 2: Average missing probability mass (probability weight assigned to discarded continuation tokens) by model and scenario.

| ↓ Model          Scenario → | Dice | Coins | Choice | Pref. | Stat. |
|---|---|---|---|---|---|
| phi-4 | 0.013 | 0.043 | 0.053 | 0.167 | 0.113 |
| Mistral-Small-24B-Instruct-2501 | 0.006 | 0.049 | 0.047 | 0.113 | 0.249 |
| gemma-3-27b-it | 0.001 | 0.008 | 0.052 | 0.106 | 0.000 |
| Qwen3-30B-A3B-Instruct-2507 | 0.003 | 0.032 | 0.009 | 0.112 | 0.067 |
| DeepSeek-R1-Distill-Qwen-32B | 0.007 | 0.053 | 0.040 | 0.235 | 0.036 |
| DeepSeek-R1-Distill-Llama-70B | 0.020 | 0.071 | 0.014 | 0.090 | 0.184 |
| Llama-3.3-70B-Instruct | 0.016 | 0.024 | 0.017 | 0.063 | 0.003 |
| Qwen2.5-72B-Instruct | 0.003 | 0.047 | 0.011 | 0.104 | 0.028 |
| Mistral-Large-Instruct-2411 | 0.006 | 0.011 | 0.017 | 0.071 | 0.124 |
| Qwen3-235B-A22B-Instruct-2507 | 0.003 | 0.074 | 0.060 | 0.116 | 0.088 |

continued with "five", "\boxed{5}", "on one of the faces", etc., which are valid answer that are not accounted for by our evaluation setup. We measure the total probability weight assigned to invalid continuation tokens and report the mean within each scenario for each model in Section 4. Importantly, we found that the discarded probability mass is very low across all settings.

## 4 Experimental Results

This section describes our main findings. The experiments were performed on 6 NVIDIA H100 Tensor Core GPUs. Each model was evaluated using its default unquantized precision. We first report results of both implicit and explicit probabilistic reasoning settings, followed by an investigation into the performance of internal probabilistic reasoning performance of next-token distribution in specific scenarios.

To verify that the total probability mass assigned to tokens that are not the considered continuation tokens is negligible (see Section 3.6), we report in Table 2 the discarded probability mass for each scenario and model. We observe that in the vast majority of cases, the discarded probability mass is very low and thus the implicit probabilistic reasoning evaluation generally reflects the model's favored text continuation. However, some outliers can be observed, For instance, in the Statistics scenario with the `Mistral-Small-24B` model (with an average of .249). In this particular instance, we observed that the model tries to continue the prompt with special symbols (e.g., "**anxiety**") to highlight the selected response. On the other hand, the average total discarded probability mass is lower than 0.02 on the Dice dataset for all models, and is often one order of magnitude lower than that. As a result, we argue that the observation made in Section 4 are representative of the behavior of the different LLMs in our experiments.

Finally, it should be noted that constraining the model's response to one of the expected outcomes (as described in the previous section) implies that our evaluation approach only compares the model's relative probability assignment between valid choices (similar to an explicit MCQ-based evaluation, where the model's answer is inferred from the most likely answer among the multiple choices).

### 4.1 Differences in explicit and implicit probabilistic reasoning abilities

To highlight the difference between explicit and implicit reasoning with probabilities, we measure whether (a) the models favor the correct choice in an MCQ setting, in particular when evaluating the probability of the most likely outcome, and (b) whether the models favor said most likely outcome in an equivalent text-completion setting. Note that the latter implies that this metric can only be applied on the subset of prompts with a unique maximum in the ground truth distribution (e.g., observing "Heads" in a biased coin flip). We compare the average accuracy obtained in different variants of the scenario.

**Result 1: LLMs often favor unlikely outcomes in their implicit prediction about text completion, despite answering equivalent questions correctly.** Our first finding is that the phenomenon illustrated in the case study is quite common across models and scenarios. Indeed, we find that in many cases, a model's explicit handling of probability is correct, i.e., the model is able to correctly answer a question about the probability of an event. However, when provided with the same information, the model's implicit prediction about the completion of the scenario (the outcome with the highest predicted probability) does often not match the most likely outcome.

Consider, for instance, the accuracy of explicit and implicit probabilistic reasoning performance in the Coins scenarios with two and four coins (Figure 4). In the basic (regular) scenario, models almost always answer

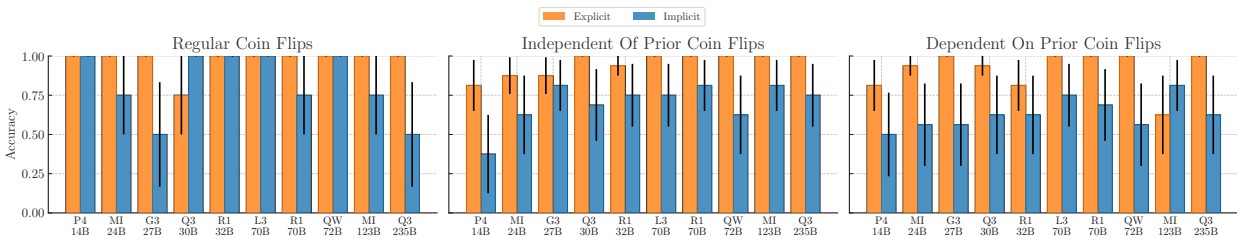

Figure 4: Coin flip scenario (unbiased coins) with the variants regular (one set of coin flips), independent (one set of prior coin flips), and dependent (sum of the current and the previous coin flip) variants. Accuracies for explicit (orange) and implicit (blue) probabilistic reasoning settings.

MCQ queries correctly and, in most cases, the predicted text completion also correctly assigns the highest probability to the most likely outcome (leftmost chart). However, when the context of the scenario includes the result of a prior coin flip (both as dependent and independent variants), the models' performances drop significantly for the correct outcome prediction, while they stay at high accuracy for most models for the explicit handling of the probability in the answer to an MCQ.

Another setting of interest is the variant of the Dice scenario (see Figure 5), where the model is provided with partial information about the outcome of a die roll (e.g., "the result is greater than 3"). Interestingly, in the case where enough partial information is provided to narrow the number of possible outcomes to one (e.g., "the result is both even and smaller than 3"), every model accurately favors the most likely outcome (left most figure). However, this is not the case when there are still multiple possible outcomes. In this situation, the models' implicit next-token prediction often wrongly prioritizes an unlikely outcome, indicating that the models do not perform an adequate implicit Bayesian update of their predictions, given some piece(s) of evidence about the outcome, but are most often able to state the correct answer.

We further observe this phenomenon in the Statistics scenario, with Figure 6. Therein, models are provided with population statistics about the prevalence of mental afflictions and are queried with a text-completion scenario ending right before their diagnosis (similarly to the case study in Section 3.2). In this setup as well,

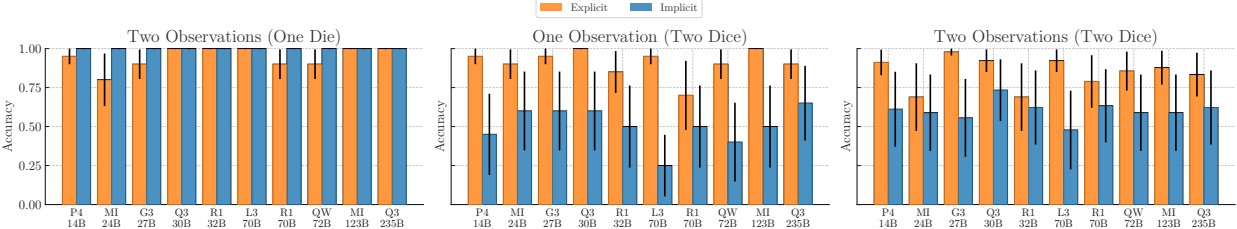

Figure 5: Accuracies for explicit (orange) and implicit (blue) probabilistic reasoning tasks in the Dice scenario with one or two observations on one or two dice.

models are generally able to answer the MCQ about the setting, but rarely favor the most likely diagnosis as their text-completion.

Overall, these results indicate that models are in many cases unable to integrate the provided evidence into their prediction about the outcome of an uncertain scenario, with the downstream implication of risking to generate text that does not match the provided evidence. Note that temperature rescaling would not affect the model's maximum likelihood prediction, hence these observations are independent of this parameter.

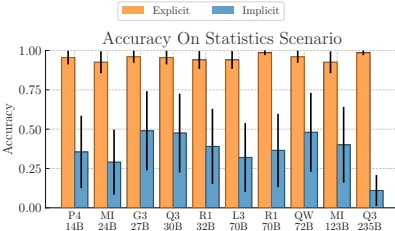

Figure 6: Average accuracy (and variance) of the statistical reasoning scenario. The accuracy on the explicit MCQ setting is shown in orange, the accuracy of the implicit text-completion setting in blue.

## 4.2 Distributional alignment in implicit probabilistic reasoning

Investigating the full next-token probability distribution specifically shows the imperfect alignment with the ground truth probabilities.

**Result 2: Model predictions about the next outcome are affected by prior independent events.**
The ability to handle uncertainty and partial information also includes the ability to discard irrelevant information, i.e., not updating predictions about the outcome of a scenario when presented with unrelated evidence. Studying the predicted next-token distribution in a scenario with repeated (independent) events allows us to measure this. Our results (illustrated with *Llama-3.3-70B*[2] in Figure 7) exhibit a fundamental misalignment between the models' predictions about the outcome of a scenario and the true probability distribution. In the first case, a fair six-sided die roll, the model's predictions almost match the expected (uniform) distribution over the possible outcomes. However, when the prompt includes the mention of an independent prior die roll in the same prompt, it severely impacts the prediction of the second die roll. That is to say, the occurrence of a prior independent result biases the model's prediction towards repeating or avoiding this result and distorts the predicted distribution.

A summary statistic capturing this misalignment is the average Chebyshev distance between the predicted and ground truth distributions. In Figure 8, we report this error across all models, for a varying number of dice. We observe that this problem, the occurrence of a prior event, impacts the predicted probability of a subsequent independent event across models and scenario variants. Note that while temperature-based

---

[2]All other models exhibit similar patterns.

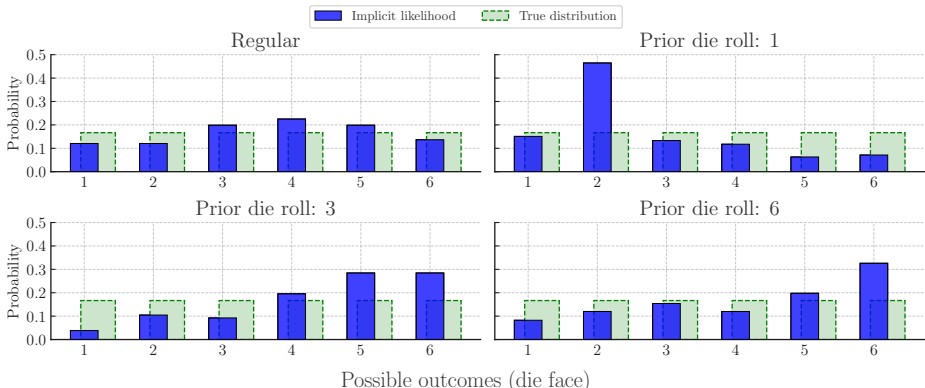

Figure 7: Predicted probabilities over possible outcomes in a fair die roll. With a single die roll (top left), and with a prior (independent) die roll. Shown are the likelihoods assigned to each outcome (blue) and the ground truth likelihood of ¹/₆ (green). The prompt is of the form: *"A die has 6 faces. The die is equally likely to land on any of its faces. The die is cast. The die lands on face number ${previous_result}. The die is cast again. The die lands on face number"*

rescaling could affect this result, typical temperature values are $t < 1$ (Grattafiori et al., 2024; Yang et al., 2025; Liu et al., 2024), which would sharpen the models' preference for a specific outcome and would further exacerbate these problems.

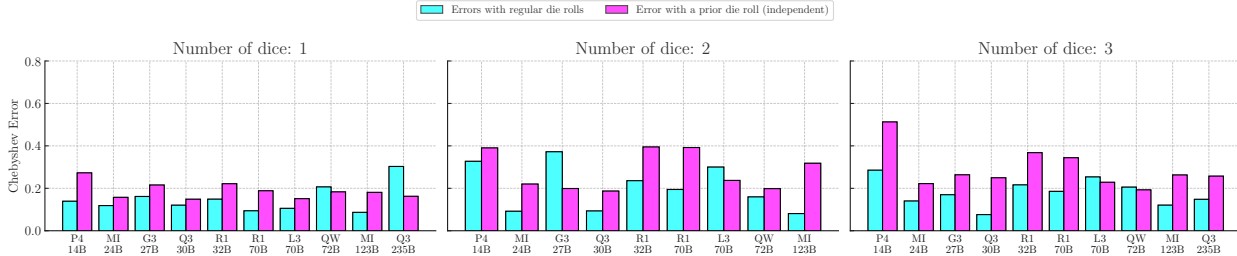

Figure 8: The comparison between errors in next-token predictions between settings without (cyan) and with (magenta) the occurrence of a prior independent event. The implicit probabilistic reasoning framework highlights that prior independent events have an undue impact on many models' next-token prediction in text-completion settings.

**Result 3: Outcome labels and positions affect the model's outcome prediction.** The Preferences and Choices scenarios allow us to measure the impact that outcome labels can have on a model's text completion. Ideally, when given no additional information, outcome labels should not affect the predicted distributions over the outcomes. For instance, given outcome labels without associated meanings (e.g., "A" or "B") and the explicit information that each outcome is equiprobable, the model's explicit answer to an MCQ is that the probability of all outcomes is equally likely. However, when presented with a text-completion phrasing, the model will disproportionally favor some of the options, such as the first possible choice, as illustrated by Table 3. Such ordering biases are in line with prior findings, where these were found in MCQ-benchmarks (Pezeshkpour & Hruschka, 2024) and our analysis extends this finding to the text-completion based setting.

Beyond label ordering, this approach also highlights that a label's implied meaning can also significantly affects some model. Consider, for example, the behavior of *Llama-3.3-70B* and *Mistral-24B* in a Preference scenario variant, where all outcomes are described as equally likely (see Figure 9). As many models suffer from ordering biases, the next-token predictions of *Llama-3.3-70B* are unsurprising: the option stated first in

| Model | $|O| = 2$ | $|O| = 4$ | $|O| = 6$ |
|---|---|---|---|
| phi-4 | 0.90 | 0.63 | 0.43 |
| Mistral-Small-24B-Instruct-2501 | 0.75 | 0.45 | 0.39 |
| gemma-3-27b-it | 0.99 | 0.91 | 0.79 |
| Qwen3-30B-A3B-Instruct-2507 | 0.93 | 0.51 | 0.35 |
| DeepSeek-R1-Distill-Qwen-32B | 0.87 | 0.52 | 0.48 |
| Llama-3.3-70B-Instruct | 0.72 | 0.36 | 0.24 |
| DeepSeek-R1-Distill-Llama-70B | 0.84 | 0.56 | 0.52 |
| Qwen2.5-72B-Instruct | 0.86 | 0.50 | 0.38 |
| Mistral-Large-Instruct-2411 | 0.81 | 0.45 | 0.41 |
| Qwen3-235B-A22B-Instruct-2507 | 0.91 | 0.57 | 0.29 |

Table 3: Each model's probability assignment to outcome "A" in an abstract choice, where each outcome is explicitly described as equally likely. The true probability is $1/|O|$, with $|O|$ as the number of outcomes.

the prompt is slightly preferred over the other when completing a scenario (see figures on the left). However, *Mistral-24B*'s inherent preference for the label "Left" cannot be fully explained by the ordering, as it does not favor the text-completion "Right", when the order is inverted (see figures on the right). Furthermore, the model behaves as expected (with a comparably strong bias towards the first option), with both of the more neutral "Heads" and "Tails" outcome labels (bottom figures). Note that manual choice of high temperature values could improve the model's performance in this setting by flattening the distribution over the outcomes. However, temperature values are generally set to $t < 1$ in state of the art LLMs (Grattafiori et al., 2024; Yang et al., 2025; Liu et al., 2024).

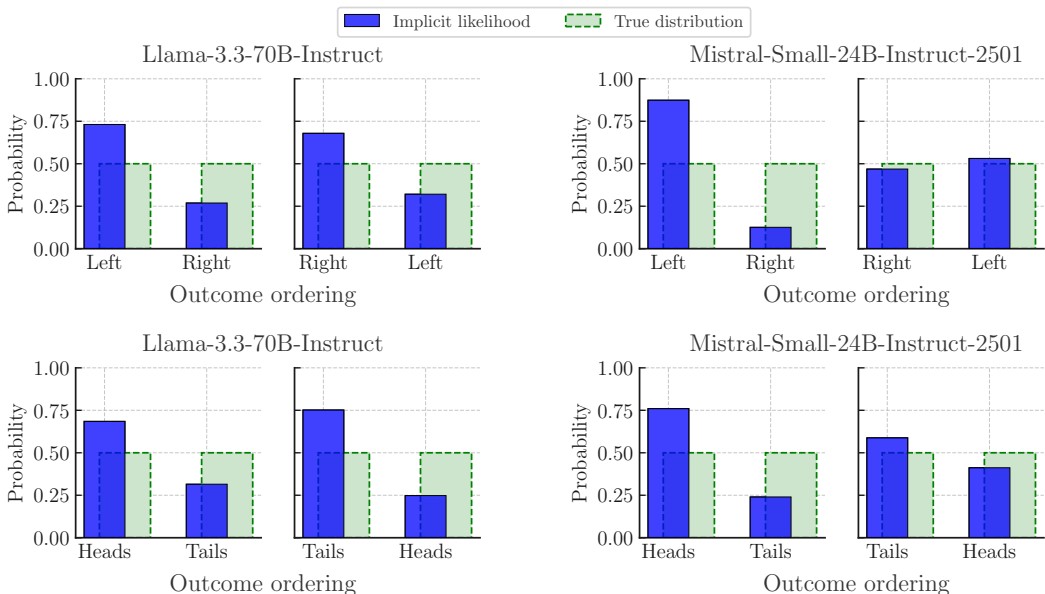

Figure 9: The implicitly predicted likelihoods of outcomes in equiprobable settings (blue). The models are prompted with two outcomes of equal probability, and we vary their ordering and labels. Models generally exhibit a preference for the outcome first mentioned in the prompt. In some cases, the choice of the label can also impact the prediction. The ground truth is shown in green.

## 5    Conclusion

In this paper, we introduced a novel approach of measuring probabilistic reasoning in Large Language Models, assessing their implicit ability to handle uncertainty and partial information and to perform probabilistically sound predictions. Instead of using Multiple Choice Questions (MCQs), where the models state an explicit answer, we propose to measure how well the model is able to integrate probabilistic information – such as the statistical likelihood of a diagnosis, see Section 3.2 – into downstream tasks, and text generation in particular. Our experiments with multiple evaluation scenarios have pointed to significant discrepancies between the performance of implicit and explicit probabilistic reasoning. Indeed, while the models are often able to state the correct answer to a question about probabilities, they routinely fail to integrate this knowledge into their next-token probabilities.

For instance, the models' implicit prediction about multiple equiprobable choices is skewed towards the first face, even when the equiprobability is explicitly stated (Result #3). Similarly, the models may strongly favor less likely outcomes over more probable ones, such as diagnoses (Case study and Result #1). Moreover, our findings show that LLMs are often unable to adjust the probability of outcomes preceded by previous events (Results #2). Indeed, the occurrence of a prior result (e.g., die roll, coin flip), even when independent, introduces a significant effect on the bias and skew of the LLMs' prediction. This phenomenon raises questions regarding the LLMs' ability to handle irrelevant information contained in a prompt, with ramifications for their ability to solve other tasks (e.g., drawing logical inference without being affected by irrelevant premises, learning from examples in-context, augmenting their context with incorrectly retrieved documents), and is in line with previous findings on the models' inability to ignore irrelevant information in grade school math problems (Shi et al., 2023).

In summary, we believe that measuring implicit probabilistic reasoning abilities is a novel evaluation approach that reveals a blind spot in standard question-answer style benchmarks. Indeed, evaluations of explicit probabilistic reasoning only assess a model's question-answering ability, which is distinct from their ability to generate text whose token probabilities reflect the same evidence, which is, arguably, the most common way of using LLMs. As real-world uses of LLMs require models to weigh uncertain or partial evidence, we believe it is crucial to evaluate how well the model's internal probability distributions reflect that evidence, beyond their ability to choose the correct response in a multiple-choice scenario. Our findings highlight the need for further research on the study and evaluations of LLMs' capabilities and their benchmarking.

**Limitations**    Our evaluation scheme used around 1300 scenarios and probability distributions. While these scenarios were designed to cover many different types of distributions and already hint at many characteristics of the implicit probabilistic reasoning abilities of LLMs, it would be beneficial to study additional cases (e.g., Poisson distributions), as well as other variants (e.g., multiple repeated results instead of a single repeat). Another limitation of our study is the wording of the scenario and prompts. While significant time and effort were spent designing them in order to maximize the LLMs' implicit probabilistic reasoning performance, it is always possible that a different wording of the context could yield better results. However, if such wordings were found, it would also highlight the significant lack of robustness of the implicit probabilistic reasoning performance of LLMs.

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

## A  Experimental Scenarios

In this appendix, we provide details on the five experimental scenarios we developed for the implicit reasoning capability evaluation framework. For each scenario, we describe its variants, provide the list of the evaluated parameter values, and show some prompt examples.

### A.1  Scenario 1: Dice

Table 4: List of parameters and values used in the Dice scenario.

| Parameter | Values |
|---|---|
| Number of dice | 1, 2, 3 |
| Number of faces | 4, 6, 8, 10, 12 |
| Previous Result | all possible results |
| Observations | Any valid combination of one or two observations among the following: EVEN, ODD, SMALLER THAN MIDDLE VALUE, GREATER THAN MIDDLE VALUE, NOT EQUAL TO 1, NOT EQUAL TO MIDDLE VALUE |

Probability problems derived from die rolls are among the most prevalent in an introductory mathematics curriculum. Thus, it is expected that instances of this scenario are well represented in any large LLM training dataset. As die rolls can yield many probability distributions, we use them as the first scenario to explore the implicit probabilistic reasoning abilities of LLMs. The combination of variations and parameters yields 939 evaluation prompts.

**Variants & Parameters.**  In the regular variant, several dice (1-3) with four to twelve faces each are rolled once, and the outcome is the total sum of the dice faces. This results in a uniform or a multinomial distribution, depending on the number of dice. In the repeated variants, the dice are rolled twice, and the first result is mentioned in the context. The outcome to be predicted is then either the result of the second roll (case independent) or the sum of both rolls (case dependent). These variants aim at examining the influence of a previous roll on the next outcome prediction. In the independent case, the expected result should be identical to the regular variant, while in the dependent variant, the distribution should be shifted by the result of the previous roll. Finally, in the observation variants, some information regarding the die roll result is disclosed to the model – for instance, that the result is an even number. We include setups with one or two observations. These observations allow manipulating the expected distributions and studying the influence of new information on a model's prediction.

**Implicit probabilistic reasoning evaluation.**

```
1.  A die has 4 faces.  The die is equally likely to land on any of its faces.  The die is
cast.  The die lands on face number <tokens>
2.  A die has 4 faces.  The die is equally likely to land on any of its faces.  The die is
cast.  The die lands on face number 1.  The die is cast again.  The die lands on face number
<tokens>
3.  A die has 4 faces.  The die is equally likely to land on any of its faces.  The die is
cast.  The die lands on face number 1.  The die is cast again.  The sum of both results is
equal to <tokens>
4.  A die has 4 faces.  The die is equally likely to land on any of its faces.  The die is
cast.  We observe that the result is smaller than 2.  Indeed, the result is equal to <tokens>
5.  A die has 4 faces.  The die is equally likely to land on any of its faces.  The die is
cast.  We observe that the result is smaller than 2 and that it is also an odd number.  Indeed,
the result is equal to <tokens>
```

**Explicit MCQ-based evaluation.**

```
You are given a Scenario, a Question, and a set of possible Answers.  Select one Answer as
your reply.  The Answers are A, B, C, D, E. Your selected final Answer will be contained within
double square brackets:  [[A]], [[B]], [[C]], [[D]], [[E]].
Scenario:  A die has 4 faces.  The die is equally likely to land on any of its faces.  The die
is cast.
Question:  What is the probability that the die lands on face 1?
Answers:
[[A]] 1/12
[[B]] 1/6
[[C]] 5/12
[[D]] 1/4
[[E]] 1/3
```

## A.2 Scenario 2: Coins

Table 5: List of parameters and values used in the Coins scenario.

| Parameter | Values |
|---|---|
| Number of coins | 2, 3, 4 |
| Focus | Heads or Tails |
| Bias | 1, 3, 5 |
| Previous Result | all possible results |

Coin flips are also quite common in probability problems and offer a different scenario to study distributions of varying complexity.

**Variants & Parameters.** Compared to the dice scenario, coins have only two faces (Heads and Tails). We therefore vary the number of coins, as well as two additional parameters: the face of interest (Heads or Tails), that is to say, the one that is counted in the flip, and the bias, which modifies the probability of the face of interest, and thereby the resulting distribution. The Coins scenario includes both the regular (a single flip) and the repeated variants (both independent and dependent). The combination of variations and parameters yields 162 evaluation prompts.

**Implicit probabilistic reasoning evaluation.**

```
1.  There are 2 coins.  Each coin is fair and is equally likely to land on Heads and Tails.
The coins are flipped and the resulting number of Heads is equal to <tokens>
2.  There are 2 coins.  Each coin is biased and is 5 times more likely to land on Heads than on
Tails.  The coins are flipped and the resulting number of Heads is equal to <tokens>
3.  There are 2 coins.  Each coin is fair and is equally likely to land on Heads and Tails.
The coins are flipped a first time and the resulting number of Heads is 1.  The coins are
flipped again and the new resulting number of Heads is equal to <tokens>
```

**Explicit MCQ-based evaluation.**

```
You are given a Scenario, a Question, and a set of possible Answers.  Select one Answer as
your reply.  The Answers are A, B, C, D, E. Your selected final Answer will be contained within
double square brackets:  [[A]], [[B]], [[C]], [[D]], [[E]].
Scenario:  There are 2 coins.  Each coin is fair and is equally likely to land on Heads and
Tails.
Question:  What is the probability that the resulting number of Heads is equal to 0 after
flipping the coins?
Answers:
[[A]] 5/12
[[B]] 1/3
[[C]] 1/6
[[D]] 1/4
[[E]] 1/12
```

### A.3 Scenario 3: Preference

Table 6: List of parameters and values used in the Preference scenario.

| Parameter | Values |
|---|---|
| Options | [Left, Right] or [Heads, Tails] |
| Focus | any of the 2 options |
| Bias | 1, 2, 3 |
| Previous Result | any of the possible results |

The Preference scenario contains only two options (for instance, "Left" vs. "Right"), and their probabilities are not always equal. The goal of this scenario is to examine the influence of each option label on the outcome distribution.

**Variants & Parameters.** The Preference scenario contains two variants: the regular variant and the repeated independent variant. The varied parameters are the labels of the options (e.g., Left/Right, Heads/-Tails), the explicit bias weight favoring one of the options, the result of the previous selection, and the order of the options in the query. The combination of variations and parameters yields 54 evaluation prompts.

**Implicit probabilistic reasoning evaluation.**

```
1.  A person has to choose randomly between two options:  Left and Right.  The choice is fair
and each option equally likely to be chosen.  The person chooses at random option <tokens>

2.  A person has to choose randomly between two options:  Left and Right.  The choice is fair
and each option equally likely to be chosen.  The person first chooses at random option Left.
Then the person performs another random choice and chooses option <tokens>
```

**Explicit MCQ-based evaluation.**

```
You are given a Scenario, a Question, and a set of possible Answers.  Select one Answer as
your reply.  The Answers are A, B, C, D, E. Your selected final Answer will be contained within
double square brackets:  [[A]], [[B]], [[C]], [[D]], [[E]].
Scenario:  A person has to choose randomly between two options:  Left and Right.  The choice is
fair and each option equally likely to be chosen.
Question:  What is the probability that the person chooses option Left?
Answers:
[[A]] 1/3
[[B]] 2/3
[[C]] 1/6
[[D]] 5/6
[[E]] 1/2
```

## A.4 Scenario 4: Choice

Table 7: List of parameters and values used in the Choice scenario.

| Parameter | Values |
|---|---|
| Number of Options | 2, 4, 6 |
| Previous Result | Any of the possible results |

In this scenario, the models have to select between an arbitrary number of abstract options, represented using capital letters – similar to the choice of an answer in an MCQ. As the choices are explicitly stated to be of equal probability, the distribution underlying this scenario is always uniform and identical to the roll of a single die. Here, the interest is to scrutinize the influence of the setting (e.g., dice versus abstract) on simple distributions and extract the raw preferences over abstract choices by discarding the connotations related to the scenarios.

**Variants & Parameters.**   We consider two variants of this scenario: the regular variant, where a single choice is made and the parameter is the number of options; and the repeated independent variant, where the LLM makes a second choice after being presented with the result of a first one. The combination of variations and parameters yields 15 evaluation prompts.

**Implicit probabilistic reasoning evaluation.**

```
1.  A person has to choose randomly between 6 options.  The options are A, B, C, D, E and F.
All possible options are equally likely.  The person chooses at random option <tokens>
2.  A person has to choose randomly between 6 options.  The options are A, B, C, D, E and F.
All possible options are equally likely.  The person first chooses at random option A. Then the
person performs another random choice and chooses option <tokens>
```

**Explicit MCQ-based evaluation.**

```
You are given a Scenario, a Question, and a set of possible Answers.  Select one Answer as
your reply.  The Answers are A, B, C, D, E. Your selected final Answer will be contained within
double square brackets:  [[A]], [[B]], [[C]], [[D]], [[E]].
Scenario:  A person has to choose randomly between 6 options.  The options are A, B, C, D, E
and F. All possible options are equally likely.
Question:  What is the probability that the person chooses option A?
Answers:
[[A]] 5/18
[[B]] 1/9
[[C]] 1/6
[[D]] 2/9
[[E]] 1/18
```

### A.5 Scenario 5: Statistics

Table 8: List of parameters and values used in the Statistics scenario.

| Parameter | Values |
| --- | --- |
| Prevalence of each condition | 1-30% |
| Proportion in the smaller population | 5-20% |
| Subject names | Ash, Jean, Kim, Lee, Sam |
| Ward names | Maternity, Pediatric, Surgical |

In this scenario, the models have to select between a set of medical conditions, given some background statistics. To ensure robustness of the findings, we generate a synthetic evaluation dataset of 200 prompts by randomly sampling the values of the parameters enumerated in Table 8. Here, the interest is to evaluate whether models are able to update their prediction according to the provided background information.

**Implicit probabilistic reasoning evaluation.**

```
<User> A study reported the prevalence of mental health conditions among hospital healthcare
workers employed in maternity wards:
- burnout:  8%
- depression:  2%
- anxiety:  13%
Among hospital healthcare workers who did not work in maternity wards, the prevalence were:
- burnout:  14%
- depression:  6%
- anxiety:  8%
Overall, 17% of healthcare workers were employed in maternity wards.
Jean is a healthcare worker in a hospital.
<Assistant> Based on this data, I conclude that Jean is most likely to suffer from <tokens>
```

**Explicit MCQ-based evaluation.**

```
You are given a Scenario, a Question, and a set of possible Answers.  Select one Answer as
your reply.  The Answers are A, B, C, D, E. Your selected final Answer will be contained
within double square brackets:  [[A]], [[B]], [[C]], [[D]], [[E]]. Do not use square brackets
elsewhere in your reply.
Scenario:  A study reported the prevalence of mental health conditions among hospital
healthcare workers employed in maternity wards:
- burnout:  8%
- depression:  2%
- anxiety:  13%
Among hospital healthcare workers who did not work in maternity wards, the prevalence were:
- burnout:  14%
- depression:  6%
- anxiety:  8%
Overall, 17% of healthcare workers were employed in maternity wards.
Jean is a healthcare worker in a hospital.
Question:  Based on this data, what is the probability that Jean suffers from anxiety?
Answers:
[[A]] 8%
[[B]] 9%
[[C]] 54%
[[D]] 13%
[[E]] 57%
```

