# OpenReview forum: "Implicit Probabilistic Reasoning Does Not Reflect Explicit Answers in Large Language Models"
_TMLR — Accepted by TMLR_

### Review · Reviewer_XV8T · 2025-11-04

**Summary Of Contributions:**

The paper investigates implicit probabilistic reasoning in LLMs by comparing (i) performance on explicit probability MCQs and (ii) sequence-level next-token probability assignments in text-completion probabilistic tasks. Several papers analyze next-token probabilities and confidence scores for MCQs, but this manuscript’s specific focus, distribution-level alignment for probabilistic MCQs and scenarios, could be a helpful addition if the methodology is sound and the analysis digs deeper. The testing is rigorous with multiple LLMs and tasks.

**Audience:**

No

**Audience Explanation:**

The main reason the reviewer may find these results of low interest is that token probabilities have already been extensively analyzed in prior work (especially in linguistics journals) and shown to be inadequate. If such probabilities fail even on simple tasks, for instance, assigning unequal likelihoods to “left” versus “right” when describing directions, it’s unclear why they would be informative in this context. The paper would be more compelling if the authors had discovered contrary evidence or proposed a method to improve these probability estimates, rather than simply restating a well-known limitation.

**Broader Impact Concerns:**

No concern here

**Claims And Evidence:**

No

**Claims Explanation:**

The paper needs a precise, auditable explanation for mapping outcomes to model probabilities. In unconstrained completion, models can produce surface forms such as "seven", "7", "7.", "the answer is 7", "7%", or "6.0". If probabilities are only read from a single token (e.g., the token for "7"), you systematically miss mass assigned to alternate tokenizations/surface forms, especially with smaller models that won't follow the instructions well. This issue must be addressed, and the reviewer suspects it is affecting the results of experiments with a dependent variable, simply because the prompt is more complicated and the output can vary in style.

Other than this issue, the reviewer does not observe an issue with the experiments.

**Requested Changes:**

1. The author should address the issue raised about the validity of the results (Critical for Acceptance)
2. The paper could benefit from realistic datasets instead of/in addition to the simple synthetic datasets used in the paper. Refer to [1] and [2] for starting points.
3. After the technical issue is resolved, the reviewer believes the paper would benefit from a deeper analysis of the effect of the complexity of the question on the answer, instead of just two categories in Figure 8.
4. Another direction the paper can take is to analyse methods to improve the probabilities, such as the work done in [3].

[1] "What Are the Odds? Language Models Are Capable of Probabilistic Reasoning" Paruchuri et al

[2] "Extracting Probabilistic Knowledge from Large Language Models for Bayesian Network Parameterization" Nafar et al

[3] "BIRD: A Trustworthy Bayesian Inference Framework for Large Language Models" Feng et al

---

> ### Author Response · Authors · 2025-12-09
> **Response part 1**
>
> Thank you for your comments and feedback. We hope that the following answers and improvements in our paper address your concerns.
>
>
> ### Comment 1.
> The paper needs a precise, auditable explanation for mapping outcomes to model probabilities (...)  because the prompt is more complicated and the output can vary in style.”
>
>
> ### Response 1
>
> We agree that the risk of losing significant probability mass by neglecting other possible and equally valid responses (e.g; “seven” instead of “7”, etc.) may indeed lead to erroneous observations when examining the distribution only on the valid target tokens.
> However, we controlled for this phenomenon in our experiments by measuring the total missing probability mass – TMPM for short–, i.e. the total probability mass allocated by the LLM to all the tokens that were not directly measured in our experiments. This probability mass provides an upper bound on  the sum of the probability of alternate responses (e.g. “seven”).
>
> We report the values of TMPM below, and added it to our paper (see Table 1 in Section 3.5) following the reviewers comments.
>
> To measure TMPM, we define a list of allowed response keys (e.g., 1 to 6 for the simple die throw) and systematically collect the total probability assigned to other keys for all prompts. Such discarded answers include, but are not limited to, the highlighted examples (e.g. "seven", "the answer is seven") and other potential issues (e.g.,"\boxed{7}", "#7").
> It can be seen that these values are very low in the vast majority of cases, and thus cannot fully account for the significant contributor to the misalignment between observed and true distributions.
>
> | Model                           |   Dice |   Coins |   Choice |   Preference |   Statistics |
> |:--------------------------------|-------:|--------:|---------:|-------------:|-------:|
> | gemma-3-27b-it                  |  0.001 |   0.008 |    0.052 |        0.106 |  0.000 |
> | Qwen3-30B-A3B-Instruct-2507     |  0.003 |   0.032 |    0.009 |        0.112 |  0.067 |
> | Qwen2.5-72B-Instruct            |  0.003 |   0.047 |    0.011 |        0.104 |  0.028 |
> | Qwen3-235B-A22B-Instruct-2507   |  0.003 |   0.074 |    0.060 |        0.116 |  0.088 |
> | Mistral-Small-24B-Instruct-2501 |  0.006 |   0.049 |    0.047 |        0.113 |  0.249 |
> | Mistral-Large-Instruct-2411     |  0.006 |   0.011 |    0.017 |        0.071 |  0.124 |
> | DeepSeek-R1-Distill-Qwen-32B    |  0.007 |   0.053 |    0.040 |        0.235 |  0.036 |
> | phi-4                           |  0.013 |   0.043 |    0.053 |        0.167 |  0.113 |
> | Llama-3.3-70B-Instruct          |  0.016 |   0.024 |    0.017 |        0.063 |  0.003 |
> | DeepSeek-R1-Distill-Llama-70B   |  0.020 |   0.071 |    0.014 |        0.090 |  0.184 |
>
> We observe that there are outliers, such as DeepSeek-R1-Distill-Qwen-32B in the Preference scenario (with an average of .235), but they can usually be explained by the model trying to continue the prompt with special symbols (e.g.,
> “**anxiety**”) to highlight the selected response. As these exceptions occur in only rare model and scenario configurations, we believe that our downstream observations on the remaining probability mass are valid. This discussion has been added to the paper as well (Section 3.5)
>
> To answer your request for auditability, we further add a self-contained script to reproduce these measurements. The source code file (missing_mass.py) and associated prompts_for_missing_mass.csv file, is a minimal variant of our experimental framework that allows to reproduce these error measurements. Therein, all scenario prompts are evaluated on a predefined set of valid outcomes and the missing probability mass is saved for each model.
>
> We hope to have addressed this critical technical issue you raised.

---

> ### Author Response · Authors · 2025-12-09
> **Response part 2**
>
> ### Comment 2.
> “The main reason the reviewer may find these results of low interest is that token probabilities have already been extensively analyzed in prior work (especially in linguistics journals) and shown to be inadequate.”
>
> ### Response 2.
> Our focus in this work lies on highlighting the discrepancy between a model’s capability to answer knowledge questions on probabilistic reasoning scenarios and its ability to integrate this knowledge into its next-token prediction. As this next-token prediction mechanism is the fundamental mechanism for LLM’s text-generation, we believe that this evaluation method reveals a capability gap that common evaluation methods do not cover yet and, to the best of our knowledge, this distinction has not been investigated before.

---

> ### Author Response · Authors · 2025-12-09
> **Response part 3**
>
> ### Critique 3.
> Realistic datasets from [1] and [2] and improvements in probabilistic reasoning in [3].
>
> ### Response 3
> Thank you for providing these references. We carefully read them and reached the following conclusions.
>
> The dataset used in [1] rigorously evaluates the models ability to answer questions about a provided probability distribution (estimating percentiles, drawing samples, and calculating probabilities). Consider the example from their dataset: “Your task is to estimate the percentile of a given annual electricity cost within the population [...]. Answer with just a numerical response from 0 to 100. Make sure your answer is enclosed by xml tags <answer> and </answer>. ## Consider the following parameters that describe a normal distribution of this data: Mean: 1792.453 Standard Deviation: 997.107 ## Here is your question: Question: What is the percentile of an annual electricity cost of $360.0? [...] Answer:”
> These evaluation prompts are conceived to evaluate models in an explicit probabilistic reasoning setting, with single correct/false answers. This makes the dataset ill-suited for our framework, as we rely on a distribution of possible true answers our evaluation setup in order to measure the models’ next-token distribution alignment with multiple outcomes of varying probability.
>
> On the other hand the dataset used in [2] could indeed be adapted for our framework. It consists of a set of 80 Bayesian networks with 5 to 50 nodes each, with the information on the probabilities of the node's states, given its parent states. These 80 networks are derived from real-world datasets (Leonelli, M. (2025). bnRep: A repository of Bayesian networks from the academic literature. Neurocomputing).
> As each of these large Bayesian networks poses a much more complicated evaluation setting than the scenarios we designed, we do not expect the inclusion of such more complex scenarios to reveal better model performance than on the simpler settings we examined in this work.
> To verify this, we adapted five of the Bayesian networks into our implicit probabilistic reasoning evaluation framework as preliminary results. Each prompt contains a textual description of the entire Bayesian network’s variables and states, and provides a hypothetical scenario (e.g., “Given the conditions zfygpa=<=0, zgpa=<=0, ugpa=<3.3, tier=1, predict the outcome for variable fam_inc.”) We construct the prompts by covering the full set of variable states to generate 171 prompts.
> In the table below, we report the results for one of the generally better performing models on our benchmark (Llama3.3-70B), on these new scenarios. We consider a response correct (to derive the overall accuracy), whenever the most likely outcome is assigned the highest probability.
> |                                         | Accuracy | Num. prompts |
> | :-------------------------------------- | -------: | ----------: |
> | BN: cardiovascular, var.: EducationLevel   | 0.278 |          18 |
> | BN: lawschool, variable: fam_inc           | 0.424 |         144 |
> | BN: nanomaterials1, var: ClassificationGHS | 0.333 |           3 |
> | BN: nanomaterials2, var.: ClassificationGHS|     0 |           3 |
> | BN: dioxins, var.: trimester               | 0.333 |           3 |
>
>
>
> We can observe that the model performs poorly on this dataset, and thus we do not believe that including it in our paper would bring additional information regarding the discrepancy between implicit and explicit reasoning. However, we would be happy to add these results to the paper if requested by the reviewer.
>
> Finally, the BIRD approach proposed in [3] is a sophisticated method to improve LLM’s probabilistic question-answering ability by offloading probabilistic reasoning to an external Bayesian model. Under this approach, the LLM is used to extract relevant factors from a natural language decision making scenario and to assign probabilities to two outcomes, given these factors, well showcased in their Fig.10 on p. 27. As this setup to estimate probabilities is fundamentally different than the implicit and explicit approaches we compare in our work, we do not believe the BIRD framework to be applicable to the issue we measure in our work. In general, the models know (i.e., are able to state the information in an answer) but do not always adapt their next-token predictions accordingly. Our goal in this work is to propose a method to measure this discrepancy in LLMs, solving this misalignment is the target of future work.
>
> We updated the Related work section to include a discussion of all these works.

---

> ### Author Response · Authors · 2025-12-09
> **Response part 4**
>
> ### Comment 4.
>
> “After the technical issue is resolved, the reviewer believes the paper would benefit from a deeper analysis of the effect of the complexity of the question on the answer, instead of just two categories in Figure 8. “
>
> ### Response 4.
> Please note that the configurations shown in Figure 8 currently distinguish three categories of complexity in the evaluation setting by varying the number of dice that are cast. We report two scores by level of complexity for each model to illustrate the difference in scores when an independent prior event has occurred and when one hasn’t. We believe that adding further results in the figure could distract from this finding, but are willing to add additional figures if requested by the reviewer.

---

> > ### Comment · Reviewer_XV8T · 2025-12-11
> >
> > The reviewer thanks the authors for the detailed and comprehensive response.
> >
> > The authors have adequately addressed the critical concern regarding tokenization and missing probability mass.
> >
> > The effort to test the Bayesian Networks dataset is appreciated. While the reviewer maintains that the reliance on simple synthetic tasks limits the paper's broader insights, it is acknowledged that, given the poor performance of open-source LLMs on realistic datasets, this approach is necessary for the current study.
> >
> > These points will be taken into account in the final recommendation.

---

### Review · Reviewer_G53Z · 2025-11-21

**Summary Of Contributions:**

The contribution of the paper lies in evaluating the difference between the MCQ style probability estimation and the token probability style of evaluation.

The authors evaluate the prediction accuracy difference between the two approaches across different scenarios, with variants control.

The authors also show the influence of the prior on the implicit reasoning evaluation.

**Audience:**

Yes

**Audience Explanation:**

Yes, evaluating the probability estimation of LLM is an important and interesting problem for people care about robustness of AI system.

**Broader Impact Concerns:**

No Concerns

**Claims And Evidence:**

Yes

**Claims Explanation:**

Yes, the paper clearly supported their claim on the mismatch between two ways of measuring probabilisitic reasoning in LLMs.

This is done through comparing the two methods across divers model scales, families, and tasks, with controllable variants and statistical error bars.

**Requested Changes:**

I would recommend the authors to add discussion on why we need to do the implicit evaluation, and why the implicit evaluation reflects the model's reasoning on probability.

When we look at the next token probability, it has a very large vocabulary, and all the irrelevant tokens will have an influence on the token of interest.  Why this is considered as the true belief of the model? What's the advantage of doing implicit evaluation?

---

> ### Author Response · Authors · 2025-12-09
> **Response to reviewer G53Z**
>
> Thank you for your feedback and comments.
>
> ### The importance of the evaluation of implicit reasoning
>
> By evaluating the model’s next-token probability assignment, our approach measures the fundamental mechanism by which LLMs generate text. In a Bayesian sense, this probability assignment could be interpreted as a credence or a model’s belief about the outcome of a scenario. In particular, this framework makes possible to measure the impact of  how probability assignments are impacted by  additional information (e.g., a prior die roll, or partial observations). We argue that generating text in accordance with a known probability reveals a model’s implicit probabilistic reasoning ability and should be measured.
> Indeed, instead of asking the model what information it knows, we measure how well the model is able to integrate this information (e.g., the statistical likelihood of a diagnosis) into its predicted distribution about the likely outcomes of a scenario.
> As illustrated by the case study (Section 3.2), a biased implicit reasoning process can lead to a skewed probability distribution over the possible outcomes, which can then lead to incorrect conclusions and misleading justifications.
> Furthermore, the model's explicitly stated answer is not necessarily indicative of a model's implicit prediction about the outcome, as the model appears able to correctly answer an MCQ without being able to integrate the information into next-token probabilities.
> Importantly, this fundamental discrepancy, as well as the biases of the implicit probabilistic reasoning process, are not specific to this case study, as it is reflected in many of our experimental results (see Section 4).
>
> We added this discussion to the Section 3.2 (last paragraph) to better explain the importance of measuring the implicit probability reasoning.
>
> ### Next token probability
> Concerning the influence of assigning probability mass to irrelevant tokens, please refer to our detailed response 1 to review RXV8T) .

---

### Review · Reviewer_PwrF · 2025-12-01

**Summary Of Contributions:**

The paper focuses on the evaluation of LLMs in answering probabilistic questions. The standard method for doing so is to ask a model to select the probability of an outcome event in a multiple-choice format. In contrast, the authors propose to rephrase the question as a partially completed text whose natural continuation is the outcome event and compare the probability assigned by the model to the respective next token with the ground truth probability. The paper introduces the evaluation methodology in Section 3 using two simple examples, and then compares the results of the two evaluation methods (multiple choice \& the authors' method) using multiple LLMs across a set of tasks that require probabilistic reasoning.

**Audience:**

Yes

**Audience Explanation:**

Since LLM evaluation is pretty much an open topic, I believe this paper could be of interest to the community. The authors put emphasis on understanding the next-token distributions induced by LLMs rather than simply evaluating their answers to multiple-choice questions, which goes in a less conventional direction within the literature on LLM evaluation. Although, as I wrote above, the main message of the paper is not as sharp as it could be, the results could be useful to other researchers in the area.

**Claims And Evidence:**

No

**Claims Explanation:**

I found the main idea of the paper intriguing, i.e., seeing if there is a mismatch between a model's chosen answers in multiple-choice probabilistic questions and the actual probabilities it assigns to the respective tokens when prompted to autocomplete a similar text, rather than directly being asked. The experiments that the authors perform to showcase that this mismatch exists seem quite comprehensive as well since they are based on multiple state-of-the-art models. The writing is clear as well, and the paper is easy to follow. However, I also believe the paper has some weaknesses which I elaborate on below:
* In its current version, the paper lacks a clear message. While it effectively shows that the evaluation method introduced by the authors yields significantly different results from standard multiple-choice, the paper neither helps us understand why that is the case nor makes a strong case why one should use the proposed evaluation method instead. Especially regarding the latter, the authors argue that an LLM's good performance on multiple-choice questions is not representative of its abilities at handling uncertainty (page 3 top), implying that their method is more representative. I was expecting a comprehensive discussion around why, conceptually, the proposed method is a better measuring device than multiple-choice questions but that seems to be missing from the paper and is being taken for granted instead.
* The paper is missing a discussion of the potential effects of reinforcement learning using human feedback (RLHF) on next-token distributions. The authors build upon the premise that an LLM, when prompted with a partial text of a probabilistic question and has to generate an outcome-token, its next-token distribution should match the ground truth distribution (i.e., be calibrated) or at least their modes should match. It is unclear why one would expect this to be true or would even want this to be true; LLMs are (pre-)trained to match the frequencies of tokens on vast text from the internet which could already introduce biases from contexts that are irrelevant to probabilistic questions. Further, text-completion LLMs are trained via RLHF to follow and respond to instruction prompts, being optimized based on human preferences to satisfy other desiderata such as helpfulness, safety, etc. It seems natural that such a process would make next-token distributions even less calibrated.
* The proposed method builds a distribution over outcomes by focusing on the next-token probabilities that the model assigns to a small subset of tokens that correspond to the outcomes of interest. To make sure that this leads to a valid distribution, the probabilities of the subset of tokens are rescaled to sum up to 1. However, it is unclear how much probability mass is lost by filtering out tokens that are not considered relevant and what effect this has on the model's performance. To be precise, consider the example of Figure 1. The authors give to the model a context "Two fair six-sided dice land on $<\text{tokens}>$" and isolate and rescale the probabilities that the model assigns to tokens "2", "3", ..., "12". I believe one would be in a position to come to a completely different conclusion about the model's abilities knowing that the total probability of those 11 tokens is almost 1 vs. if it sums up to something like $0.7$ and the rest of the mass is assigned to other completions (e.g., "[...] land on the table. Their sum is 8."). In other words, I am concerned that the conclusions reached by using the proposed method can be easily biased by the choice of the subset of tokens used as outcomes, and reporting how much probability mass falls outside those tokens is essential to properly contextualize the results.
* In addition to the points raised above, I believe the proposed method has limitations that the authors should try to address or at least acknowledge and point out to the reader. Most crucially, LLMs that generate text stochastically use a *temperature* parameter to turn the (unbounded) logits of the neural network into a valid probability distribution. Although the mode of the distribution does not change as one varies the temperature, how flat or concentrated the distribution is does change, which would significantly impact all three metrics the authors refer to in their setup in page 7 (Chebyshev, Manhattan, and Kullback-Leibler). I was expecting the authors to propose some principled way of setting the value of the model's temperature before using their evaluation method, however, the word temperature is nowhere mentioned in the paper.
* Another limitation is that the proposed evaluation method seems to be constrained to the evaluation of probabilistic questions whose outcomes can be expressed as a *single* token. While this may be ok for the simple coin tossing and dice rolling settings studied in the paper, I don't see how it could be generalized to more complex settings or in non-English languages in which the outcomes of interest can't be expressed as a single token.

**Requested Changes:**

Here, I summarize my main questions that I believe focus on major points the authors need to address in their rebuttal and in the revised version of the paper:
* Is there a clear benefit in using the proposed evaluation method instead of multiple-choice? Can the authors present some arguments clarifying why they think their method is more representative of a model's abilities to do probabilistic reasoning?
* How much probability mass is lost during the normalization of the probabilities of the subset for it to sum to 1? Can the authors report those numbers in a few (if not all) of their experiments?
* Could the authors include experiments with base models (trained to auto-complete) text in addition to the instruction-tuned models they currently evaluate? It would be insightful to see if the base models yield better calibrated next-token distributions.
* Can the authors clarify how they choose the value of the temperature parameter in their experiments? Since different models have different recommended values (usually around 0.6-0.8), how does this choice impact the results?
* Can the authors propose a way to handle outcomes that consist of more than one token? An experiment that may be helpful would be to compare a model's performance at a task (i) using the proposed method vs. (ii) letting the model generate in an unconstrained manner (i.e., without limiting to the small set of hardcoded outcomes) and then using an LLM-as-a-judge to classify the freely generated text into the set of outcomes. That would allow to see if the low performance of a model based on the proposed method is really a lack of capability or a potential artifact of the method itself.

---

> ### Author Response · Authors · 2025-12-09
> **Response  part 1**
>
> Thank you for your comments and suggestions. We hope that the responses and revisions provided in our manuscript adequately address your concerns.
>
>
> ### Comment 1.
>
>  “Is there a clear benefit in using the proposed evaluation method instead of multiple-choice? Can the authors present some arguments clarifying why they think their method is more representative of a model's abilities to do probabilistic reasoning?“
>
> ### Response 1.
>
>  Our core claim is that the proposed evaluation method is not necessarily more representative than classical evaluation methods of probabilistic reasoning, but rather that the approach unveils another (complementary) perspective in the assessment of this ability. Analogous to the difference between retrieving knowledge when prompted with a question and integrating the same knowledge into a decision, the implicit probabilistic reasoning evaluation approach aims to reveal the difference between answering questions on probability and generating text in accordance with them. Please see the last paragraph of Section 3.2 for a more detailed exploration of the importance of both approaches.
>
> ### Comment 2.
>  “How much probability mass is lost during the normalization of the probabilities of the subset for it to sum to 1? Can the authors report those numbers in a few (if not all) of their experiments?”
>
> ### Response 2.
>  We agree that the missed probability mass is an important part of the evaluation method and should be explained in the paper. Please refer to our detailed response 1 to review RXV8T for the numbers for all of the experiments.
>
>
> ### Comment 3.
> “Could the authors include experiments with base models (trained to auto-complete) text in addition to the instruction-tuned models they currently evaluate? It would be insightful to see if the base models yield better calibrated next-token distributions.”
>
> ### Response 3.
> We did not include this experiment in the current version of the manuscript for the following reasons:
> First, we wanted to ensure that the one-to-one comparison between the implicit and the explicit setting is valid. As the explicit setting relies on MCQ-style question-answering, this mode requires instruction-fine tuning. Thus, we prefer evaluating the implicit setting in this mode as well, so as to make sure that difference in performance under the two evaluation settings is not caused by the differently trained models.
> Second, as instruction-fine tuned models use the same next-token probability-based prediction for text-generation, we believe it is also useful to evaluate this setting.
> In addition, only a small portion of open-weight models are released as both base and instruction fine-tuned variants, thus excluding models where the former is available, would severely restrict the generalizability of our findings.
>
> Nonetheless, we agree that it is useful to check whether base models outperform their instruction-fine tuned variants in implicit probabilistic reasoning. Thus, we added three models (for which a base variant was available) and evaluated them on our most complex evaluation scenario (dice).
> We compare first the missed mass (i.e., probability scores assigned to discarded tokens) and the accuracies over all dice subscenario prompts (i.e., the 441 prompts where the distribution has an individual maximum). These scores hint that there is no significant difference in ability between the variants.
>
> | Model | Misallocated probability mass |
> | :-------------------------------------- | -------: |
> | Mistral-Small-24B-Base-2501 | 0.011 |
> | Mistral-Small-24B-Instruct-2501 |  0.006 |
> | Qwen2.5-72B |  0.007 |
> | Qwen2.5-72B-Instruct |  0.003 |
> | gemma-3-27b-pt |  0.009 |
> | Gemma-3-27b-it |  0.001 |
>
> |  Model | Accuracy Base |  Accuracy IF |
> | :-------------------------------------- | -------: | ----------: |
> | Mistral-Small-24B |   0.365 |   0.363 |
> | Qwen2.5-72B |  0.478 |   0.447 |
> | gemma-3-27b |  0.41 |   0.395 |
>
> Due to the aforementioned reasons, and the negligible differences in scores, we believe it is representative to evaluate instruction fine-tuned models in this framework. However, we are willing to include these results in the manuscript if requested by the reviewer.

---

> ### Author Response · Authors · 2025-12-09
> **Response part 2**
>
> ### Comment 4.
> “Can the authors clarify how they choose the value of the temperature parameter in their experiments?”
>
> ### Response 4.
> To compute the probability of the outcomes, we apply a softmax operation over the logits produced by the model, without applying a temperature-based rescaling (i.e. a temperature of 1). We do so to avoid affecting the raw probability the model produces, independently of the subsequent sampling strategy. As our further comparisons between explicit and implicit evaluation mainly rely on the maximum likelihood of the distribution (which is unchanged by the temperature), we do not believe this to significantly affect our findings.
>
> ### Comment 5.
> “Can the authors propose a way to handle outcomes that consist of more than one token?“
>
> ### Response 5.
>  Please note that we already handle outcomes that consist of more than one token in the current manuscript (e.g; “de|pression” or  “T|ails”) – we clarified this in Section 3.1; paragraph “implicit probabilistic reasoning”. We handle multiple tokens  by evaluating the joint probability of the sequence of tokens, as it represents the likelihood of generating this specific outcome. For this reason, the outcome tokens may differ between models (e.g., if “Tails” is tokenized into one or two tokens). To measure the probability of the outcome [“_T”. “ails”], we first compute the probability of p(“_T” | prompt) and multiply it with the probability assigned to p(“ails” | prompt + “_T”). This setup is handled by our framework automatically for each model. Concerning an LLL-as-a-Judge evaluation, we agree that in (future) more complex evaluation scenarios, this would be the required approach. However, as the complexity of our scenarios is intentionally designed to examine a finite set of outcomes with known probabilities, which the models directly give us, we believe that the likelihood-based evaluation is simpler and measure this ability in a simple way.

---

> > ### Comment · Reviewer_PwrF · 2025-12-12
> > **Thank you for the rebuttal**
> >
> > I would like to thank the authors for the detailed rebuttal. I appreciate the addition of Table 1 in the paper, as the amount of probability missing before re-scaling the probabilities of the tokens of interest is crucial for properly interpreting the results. I believe the rebuttal has addressed most of my concerns. Below are some follow-up comments on specific points, but they are just recommendations for the final version of the paper, so the authors shouldn't feel the need to respond.
> >
> > **[Comment 1]** I will not insist on this and I am willing to let it for the readers to judge, but I am still not entirely convinced why the proposed method is a useful measuring device, that is, why it tells us something about a model's probabilistic reasoning capabilities that multiple-choice questions (MCQs) don't tell. MCQs are a natural measuring device for LLMs because they are also used to evaluate humans in multiple settings, including probability classes in school. They may not be direct evidence of a human's (or LLM's) capacity for probabilistic reasoning but they are a good proxy, since good performance in a well-designed evaluation based on MCQs is highly unlikely, unless one memorizes the answers. On the other hand, measuring whether the next-token distribution of an LLM is well-calibrated with a ground-truth outcome distribution is (i) also simply a proxy, and (ii) somewhat unnatural. Regarding (i), directly measuring a model's capacity for probabilistic reasoning would require identifying circuits in its neural network that perform specific operations (e.g., implement Bayes theorem), so analyzing only its next-token distributions can only serve as a proxy. Regarding (ii), the human analogue would be to measure a human's ability to generate random numbers (e.g., $0/1$ with $50-50$ chances for coin tossing), which is very likely to yield a biased/uncalibrated distribution [1]. Therefore, it is unclear why a biased/uncalibrated next-token distribution of an LLM trained on human-generated text tells us something useful about its probabilistic reasoning capabilities. Again, this point will not play a major role in my evaluation of the paper, I just want to highlight that the authors could significantly expand and contextualize their discussion on "the importance of measuring implicit probabilistic reasoning abilities", which I currently find somewhat shallow.
> >
> > [1] Nickerson, Raymond S. "The production and perception of randomness." Psychological review 109.2 (2002): 330.
> >
> > **[Comment 3]** I appreciate the experiments with base models. Given the small differences in accuracy, I do not think they need to be included in the final version.
> >
> > **[Comment 4]** If I am not missing something, I believe the results based on the Chebyshev distance in Figure 8 *do* depend on the choice of temperature. I think using temperature=1 for the softmax operation is fine and does not change the main message of each experiment. However, I would strongly encourage the authors to explicitly mention in the paper that they set temperature=1. Although the method does not require sampling, it seems quite strange to not even mention it, since it is the most important parameter shaping next-token distributions.

---

### Decision · Action_Editor_9fZ7 · 2026-01-21

**Recommendation:** Accept as is

**Additional Comments:**

We have three reviewers for this paper, all of them are **Leaning Accept**.

The reviewers raised several good points during the reviews, and the authors, in turn, addressed most of them by including new experiments (ex: computing the total missing probability mass in Table 1) or adding some new discussions to the paper (ex: end of Section 3.2).

Despite that, the reviewers are not fully convinced about the usefulness and significance of the problem studied by this paper. Of course, this is a bit subjective, and none of them considered this as a reason not to recommend acceptance. Henceforth, I also recommend the acceptance of this paper. We will see how the community will benefit from this paper.

I suggest the authors to work on how they convey the message of the paper better. For instance, they added a note on the importance of implicit probabilistic reasoning abilities in Section 3.2. Perhaps parts of that message can be brought up earlier in the Introduction of the paper.

**Audience:**

Yes

**Audience Explanation:**

Yes, this paper closely studies LLMs and their inner workings, which is of interest to a large fraction of the community.

**Claims And Evidence:**

Yes

**Claims Explanation:**

Yes, all reviewers agree that the claims are clear and convincing.